# Elevated Arterial Blood Pressure as a Delayed Complication Following COVID-19—A Narrative Review

**DOI:** 10.3390/ijms25031837

**Published:** 2024-02-02

**Authors:** Emilia Bielecka, Piotr Sielatycki, Paulina Pietraszko, Agnieszka Zapora-Kurel, Edyta Zbroch

**Affiliations:** Department of Internal Medicine and Hypertension, Medical University of Bialystok, 15-540 Bialystok, Poland; emilka.bielecka96@gmail.com (E.B.); piotr.sielatycki@umb.edu.pl (P.S.); pietraszko270@gmail.com (P.P.); agnieszkazapora2@wp.pl (A.Z.-K.)

**Keywords:** COVID-19, SARS-CoV-2, hypertension, blood pressure, endothelial dysfunction, long covid, post covid

## Abstract

Arterial hypertension is one of the most common and significant cardiovascular risk factors. There are many well-known and identified risk factors for its development. In recent times, there has been growing concern about the potential impact of COVID-19 on the cardiovascular system and its relation to arterial hypertension. Various theories have been developed that suggest a connection between COVID-19 and elevated blood pressure. However, the precise link between SARS-CoV-2 infection and the long-term risk of developing hypertension remains insufficiently explored. Therefore, the primary objective of our study was to investigate the influence of COVID-19 infection on blood pressure elevation and the subsequent risk of developing arterial hypertension over an extended period. To accomplish this, we conducted a thorough search review of relevant papers in the PubMed and SCOPUS databases up to 3 September 2023. Our analysis encompassed a total of 30 eligible articles. Out of the 30 papers we reviewed, 19 of them provided substantial evidence showing a heightened risk of developing arterial hypertension following COVID-19 infection. Eight of the studies showed that blood pressure values increased after the infection, while three of the qualified studies did not report any notable impact of COVID-19 on blood pressure levels. The precise mechanism behind the development of hypertension after COVID-19 remains unclear, but it is suggested that endothelial injury and dysfunction of the renin–angiotensin–aldosterone system may be contributory. Additionally, changes in blood pressure following COVID-19 infection could be linked to lifestyle alterations that often occur alongside the illness. Our findings emphasize the pressing requirement for thorough research into the relationship between COVID-19 and hypertension. These insights are essential for the development of effective prevention and management approaches for individuals who have experienced COVID-19 infection.

## 1. Introduction

### 1.1. Arterial Hypertension (HTN)

Arterial hypertension is the most prevalent risk factor for cardiovascular diseases (CVD) [1]. The precise definition of arterial hypertension may slightly vary among different medical societies. As per the 2017 guidelines from the American College of Cardiology (ACC) and the American Heart Association (AHA), hypertension (HTN) is defined as having a systolic blood pressure of equal to or greater than 130 mmHg and/or a diastolic blood pressure equal to or greater than 80 mmHg, or if the individual is taking antihypertensive medications. Elevated blood pressure (BP) is defined as having a systolic blood pressure between 120 and 129 mmHg with a diastolic pressure less than 80 mmHg [2].

In contrast, The European Society of Cardiology and the European Society of Hypertension (ESC/ESH) [3], the International Society of Hypertension (ISH), and the National Institute for Health and Care Excellence (NICE) define arterial hypertension as having a systolic blood pressure of equal to or greater than 140 mmHg or a diastolic blood pressure of equal to or greater than 90 mmHg [3,4,5].

The pathogenesis of essential arterial hypertension is believed to be multifactorial and not yet completely understood. Several well-known factors play a crucial role in determining blood pressure levels, including the sympathetic nervous system, the renin–angiotensin–aldosterone system (RAAS), endothelial function and the plasma volume [6,7]. The development of hypertension is associated with various risk factors, such as age, obesity, family history, race, reduced nephron number, high-sodium diet, excessive alcohol consumption, physical inactivity, inadequate sleep, and social determinants [8]. Moreover, certain medical conditions, such as chronic kidney disease, renal artery stenosis, obstructive sleep apnea, pheochromocytoma, endocrine disorders, and the use of specific medications, can elevate blood pressure and lead to secondary arterial hypertension.

Arterial hypertension is a common risk factor for stroke, heart failure, and coronary heart disease [9].

Hence, it is of utmost importance to identify individuals at a higher risk of developing arterial hypertension in order to implement preventive measures aimed at averting the onset of cardiovascular diseases. Furthermore, an individual’s cardiovascular risk factors can influence the intensity of antihypertensive treatment [2,3,10]. Therefore, gaining a comprehensive understanding of all conditions that pose a threat to blood pressure regulation is of significant importance.

### 1.2. COVID-19 Cardiovascular Outcome

The severe acute respiratory syndrome coronavirus 2 (SARS-CoV-2) emerged in December 2019 in Wuhan, China. Its high infectivity quickly led to a global pandemic, resulting in a surge of infections known as Coronavirus Disease 2019 (COVID-19). The World Health Organization (WHO) declared a pandemic in March 2020 and announced the end of the pandemic in May 2023. The increasing number of cases raised concerns among the public and strained healthcare systems worldwide. It is widely recognized as one of the most significant global health challenges in recent decades. Over the course of more than three years, effective diagnostic and treatment algorithms have been developed along with preventive measures for acute COVID-19 infection. Thanks to the vaccines, the mortality levels of the disease have decreased along with the risk of severe illness.

Nevertheless, individuals with lingering symptoms continue to seek medical care globally, even after the acute phase of SARS-CoV-2 infection has subsided. More than 200 symptoms affecting various organ systems have been recognized as potential complications of COVID-19 [11]. Millions of people worldwide are estimated to be experiencing long COVID with the number of cases increasing every day [12]. There are numerous definitions of long COVID. The National Institute for Health and Care Excellence (NICE) defines “long COVID” as follows: “Post-COVID-19 syndrome is characterized by signs and symptoms developed during or after an infection consistent with COVID-19, continue for more than 12 weeks, and are not explained by an alternative diagnosis”. It usually presents with clusters of symptoms, often overlapping, which can fluctuate and change over time and can affect any system. Post-COVID-19 syndrome may be considered before 12 weeks while the possibility of an alternative underlying disease is also being assessed [13]. According to the World Health Organization, post-COVID condition is defined as the presence of post-COVID symptoms that persist past 3 months, last for at least 2 months, and cannot be explained by an alternative diagnosis [14].

In addition to the clinical case definitions, the term “long COVID” is commonly used to describe the signs and symptoms known to continue or develop after acute COVID-19. It includes both ongoing symptomatic COVID-19 (from 4 to 12 weeks) and post-COVID-19 syndrome (12 weeks or more) [15].

The association between SARS-CoV-2 infection and long-term cardiovascular risk continues to be the subject of ongoing investigations, encompassing the examination of both the frequency and the characteristics of long-term cardiovascular consequences.

Researchers have started to focus on complications following COVID-19. Every month, new studies emerge that demonstrate clinical correlations of symptoms with previous SARS-CoV-2 infection [16]. It is widely recognized that individuals recovering from COVID-19 infection often report chronic fatigue and reduced exercise tolerance, which can persist for several months, especially in those who required intensive care unit (ICU) admission [17,18,19,20]. Additionally, the emergence of anxiety and depression or the exacerbation of pre-existing mood disorders attributed not only to COVID-19 itself but also to the effects of quarantine and isolation during the pandemic, collectively predisposing individuals to a sedentary lifestyle and, consequently, weight gain, ultimately resulting in overweight and obesity [21].

Our objective is to examine the potential effects of COVID-19 infection on the increase in blood pressure and the subsequent risk of developing arterial hypertension in the long-term.

Recent findings indicate that long COVID may contribute to the activation of the sympathetic nervous system and the suppression of the parasympathetic system. The repercussions of these specific dysregulations may encompass an increased heart rate and elevated blood pressure [22,23,24]. Nonetheless, it remains unclear whether COVID-19-related dysautonomia is associated with the viruses’ neurotropic effects on autonomic pathways, immunological mechanisms, or the results from a cytokine storm [25,26,27].

Moreover, additional studies have indicated that patients with COVID-19, particularly those receiving treatment in intensive care units, tend to exhibit hyperreninemia along with hypernatremia, hyperchloremia, and volume overload [28]. All of the above mechanisms may lead to blood pressure elevation in individuals who previously presented a normal range of blood pressure values or worsen blood pressure values in those who previously exhibited well-controlled hypertension [29].

The exacerbation of classical risk factors for the development of arterial hypertension, such as obesity, reduced nephron number, high-sodium diet, excessive alcohol consumption, physical inactivity, insufficient sleep, and various social determinants, can also contribute to HTN development [30,31,32,33,34].

Nonetheless, the predominant scientific focus has been centered on the interaction between SARS-CoV-2 and angiotensin-converting enzyme type 2 (ACE-2). Recent studies propose that the disruption in the balance of the renin–angiotensin–aldosterone system may serve as a pivotal factor in the development of blood pressure regulation disorders. This disruption can be a consequence of endothelial dysfunction, characterized by an imbalance between vasodilatory and vasoconstrictive elements. The outcome is an increase in vasoconstriction, which leads to arterial wall stiffening and, in turn, weakened vasorelaxation [35,36].

This phenomenon can lead to various clinical complications, including atherosclerosis, thrombotic disease, acute coronary syndrome, or changes in blood pressure.

Figure 1 illustrates the potential mechanisms by which SARS-CoV-2 could induce arterial hypertension. Nevertheless, the precise pathogenic mechanism underlying all these changes has yet to be fully established, and further studies are needed to provide insights into this query.

## 2. Materials and Methods

The report of the methods used for this systematic review was in accordance with the Preferred Reporting Items for Systematic Reviews and Meta-Analyses (PRISMA) consensus statement [37]. We systematically searched the PUBMED and SCOPUS databases and manually scanned the reference lists of articles, focusing on randomized clinical trials exploring the complications of COVID-19 infection, such as elevated blood pressure and newly diagnosed arterial hypertension. The systematic review was not registered in any registry before starting the data analysis.

The final search was conducted on 3 September 2023. Only adult participants were considered, and pregnant women were excluded from this review. We included studies focused on outcomes occurring after at least 14 days since SARS-CoV-2 infection. MeSH terms included “COVID-19”, “coronavirus”, “SARS-CoV-2”, “hypertension”, “blood pressure”, “post-infection”, “after infection”, “following infection”, “recovery”, “long-term effects”, “sequelae,” “outcomes”, “complications”, “pathophysiology”, “mechanism”, ”physiological changes”, and “biological processes”. The terms were combined individually using the Boolean operator ‘AND’ or “OR” if it was needed. We applied additional automatic search criteria; namely, we included only English-language articles with valid full-text access that have been published since 2019. Duplicates were removed. Afterwards, the titles and abstracts of the remaining studies were screened by three reviewers (E.B., P.S., A.Z.-K. and P.P) in order to identify relevant articles. Disagreements were resolved by discussion between the two review authors; if no agreement could be reached, the third author (E.Z.) would decide the outcome. Finally, the selected eligible articles were fully reviewed.

### 2.1. Results

As depicted in the PRISMA flowchart presented in Figure 2, the initial search strategy yielded 4988 articles from PubMed and 1825 from Scopus. Following automatic criteria-based exclusion, 5538 records were removed. From citation screening, 40 additional articles were identified. Of these, 5 articles were unavailable, leaving 35 for abstract and full-text screening. Subsequently, a screening of titles and abstracts resulted in the selection of 45 articles for further analysis.

Six duplicates were excluded. Nine articles were excluded based on the criteria outlined in the PRISMA flowchart in Figure 2. The final selection criteria encompassed studies that demonstrated an increase in arterial blood pressure or a new diagnosis of arterial hypertension persisting for a minimum of one month following COVID-19 infection. Articles assessing blood pressure in the context of interventions that could potentially affect its values, such as orthostatic testing or physical exertion, were excluded.

In the end, 30 qualifying articles were included in this review. The complete search process is illustrated in Figure 2.

We conducted a quality analysis of the included papers, considering various factors. This included an assessment of the methodology, a consideration of study design, the criteria for participant selection, and the inclusion of a control group—defined as participants without a history of COVID-19 infection—along with their characteristics. We also scrutinized the data collection methods, definitions of observational points, and precision of the statistical analysis and reporting. Additionally, we verified study approval by ethical committees and the acquisition of informed consent. Our evaluation also involved checking whether the researchers acknowledged and addressed the study limitations. The details for all discussed papers are presented in Table 1. We recognize the importance of a thorough consideration of these aspects to ensure the reliability and credible interpretation of the study results.

The articles considered for analysis in this review encompassed a time frame extending from 2019 to 2023. Within this dataset of 30 included studies, the majority of research studies were carried out in the United States with seven articles originating from this region. Three articles were produced in China with Israel, Italy, and Turkey contributing two articles each to the body of literature. Additionally, there were studies conducted in diverse global locales, including Mexico, Egypt, Spain, The Netherlands, India, Brazil, Ukraine, Croatia, Estonia, Saudi Arabia, and Austria, that were included. One study did not specify a geographical location. Notably, two of the studies were conducted across multiple countries, reflecting international collaboration between scientists.

The collective sample sizes across these selected studies exhibited a wide spectrum, ranging from a modest 10 patients to a substantial 5,808,018 patient sample. It is pertinent to highlight that the studies based on the analysis of medical records were examined as a distinct subgroup. The studies were further categorized into four thematic groups, which encompassed articles addressing new-onset arterial hypertension subsequent to a COVID-19 infection, studies observing elevated blood pressure that did not meet the criteria for arterial hypertension, large retrospective cohort studies, and studies reporting no discernible influence of COVID-19 infection on blood pressure.

### 2.2. New Arterial Hypertension Development

A total of twelve studies offering insights into the development of arterial hypertension following a COVID-19 infection were identified [38,39,40,41,42,43,44,45,46,47,48,49].

Most of the qualified studies did not provide the exact diagnostic criteria of the newly discovered arterial hypertension, and they did not refer to any specific guidelines [39,40,41,44,45,46,48,49]. Xiong et al. [38] qualified patients with newly diagnosed high blood pressure as those who claimed to have been recently diagnosed with high blood pressure and now require blood pressure lowering drugs.

In Özcan’s [42] study, HTN was defined as a value of systolic blood pressure ≥ 140 mm Hg and diastolic blood pressure ≥ 90 mm Hg measured with the patient supine or a history of antihypertensive medication use. Akpek et al. [43] defined new onset hypertension as presenting values of ≥140 mmHg systolic BP and/or ≥90 mmHg diastolic BP in office measurements and ≥135 mmHg systolic BP and/or ≥85 mmHg diastolic BP in home BP monitoring according to the European Society of Cardiology guidelines. Vyas [47] also relied on the ESC guidelines, and the term “new onset of hypertension” was understood to represent a systolic BP greater or equal to 140 mmHg and/or diastolic BP greater or equal to 90 mmHg.

Eight studies examined the overall occurrence of clinical consequences. Notably, two studies by Ogungbe et al. [45] and Miguel Fernandez-Ortega [46] focused specifically on cardiac complications and assessed cognitive impairments, diabetes, and arterial hypertension, respectively. Akpek [43], Delalic [44], and Vyas [47] conducted targeted investigations that specifically addressed the development of arterial hypertension as a complication of COVID-19 infection.

A somewhat different approach was observed in the study conducted by Abumayyaleh et al. [48], where the development of complications over a 2.6 ± 4.6 month observation period was evaluated in individuals with diabetes compared to a control group of patients without diabetes. Interestingly, the frequency of hypertension was slightly lower in the individuals with diabetes (0.5% vs. 1.6%), but this difference did not achieve statistical significance. Boglione et al.’s study [41], on the other hand, aimed to assess complications in patients at least 6 months after hospitalization due to COVID-19. In this study, factors such as admission to the intensive care unit, length of hospitalization, and remdesivir treatment were also evaluated. In the context of arterial hypertension, the study reported 116 cases of hypertension one month after hospital discharge (25.8%), which decreased to 61 cases (14%) six months after discharge. The details regarding the methods of result assessment and key study characteristics can be found in Table 2.

Data collection in the majority of the studies was accomplished through telephone surveys [38,40,42,45,46,49] or by examining medical records [39,48]. Mei et al. [39] diagnosed hypertension in 6 individuals (0.16%), although the specific diagnostic verification method was not specified. In contrast, Sergi Ozcan [42] employed ambulatory blood pressure monitoring (ABPM) to detect four new cases of high blood pressure. Vyas et al. [47] not only assessed blood pressure but also examined laboratory parameters and chest computed tomography (CT) scans. Their findings indicated that the new onset of high blood pressure was associated with a more severe chest CT condition, steroid treatment, and higher levels of C-reactive protein (CRP) and ferritin. However, Delalic et al. [44], who reported a 16% risk of developing or worsening arterial hypertension one month after SARS-CoV-2 infection, did not provide specific information about laboratory markers or characteristic coexisting diseases within the studied group.

However, due to the absence of a control group in each research study, it is unclear whether those frequencies exceeded the expected level.

### 2.3. Retrospective Cohort Studies Based on Medical Record Databases

In the selected studies, we decided to separately extract 8 retrospective cohort studies in which data were obtained from medical record databases [50,51,52,53,54,55,56,57]. Since the patients were not examined specifically for the purpose of those studies and the researchers relied solely on the data retrieved from the patients’ medical records, we decided to analyze the conclusions from each study separately.

Their strengths lie in the relatively large populations, and a few of them even include international populations. Unfortunately, this approach also comes with several limitations. Among them is the difficulty in assessing the completeness of medical documentation, which introduces a risk of potential errors or omissions in the data.

All of these studies examined the risk of various clinical outcomes, particularly the development of arterial hypertension in individuals who had recovered from COVID-19. Remarkably, in only one out of eight eligible studies, no significant difference in the incidence of arterial hypertension between patients with a history of COVID-19 and the control group was observed. This particular study, conducted by Jennifer K et al. [55], was grounded in data provided by a large healthcare maintenance organization (HMO) in Israel, spanning from March 2020 to May 2021. It is noteworthy that the majority of the studies referred to in this context had observation periods lasting for at least 12 months. However, an exception is observed in Cohen’s study [52], where the observation period was limited to 120 days, and it specifically focused on a population aged 65 years and more. The findings relevant to this review can be found in Table 3.

### 2.4. Elevated Blood Pressure

Given the relatively brief time that has elapsed since the conclusion of the pandemic and the commencement of this systematic review, we made a deliberate choice to encompass studies that indicated an elevation in blood pressure as an outcome of COVID-19 infection even if they did not precisely meet the criteria for diagnosing arterial hypertension. It is reasonable to anticipate that arterial hypertension may manifest in this particular group of patients over an extended period of time.

In the selection of eight eligible studies, various clinical parameters were examined, encompassing blood pressure and other consequences of COVID-19 [58,60,61,62,63,64,65]. Additionally, as reported in the study by DeLorenzo et al. [60], individuals with pre-existing arterial hypertension experienced an elevation in their blood pressure, necessitating adjustments in their treatment regimens due to a loss of disease control. Similarly, Mahmoud et al. [65] noted a significant increase in median systolic and diastolic blood pressure levels. In the subgroup of 48 patients who exhibited a blood pressure increase of 10 mmHg or more from their baseline measurements taken prior to contracting COVID-19, either new antihypertensive medications were introduced or the dosage of their previously prescribed antihypertensive drugs was raised. The status of HTN before the infection, whether pre-existing or newly diagnosed, was not ascertained. Subsequent follow-up visits revealed that, after the initiation of pharmacotherapy, patients displayed lower blood pressure levels in comparison to their initial visits.

Gameil et al. [62] and Alfadda et al. [58] observed that, apart from heightened blood pressure, patients also displayed elevated levels of inflammatory markers in their biochemical blood tests.

Pertinent information regarding the included studies is comprehensively presented in Table 4. Unfortunately, our ability to verify the results presented in Tanni’s study [61] was influenced by the lack of proper documentation. It is imperative to highlight that these investigations were conducted on two discrete cohorts of patients, namely those with confirmed COVID-19 infection and those who remained unexposed to the virus.

Furthermore, we deem it important to conduct additional research concerning those subjects.

In the United States, a substantial study was undertaken amid the COVID-19 pandemic as a component of the Livongo Hypertension Program, which integrated digital blood pressure measurements. Upon the analysis of data from 72,706 participants, the findings revealed that the monthly adjusted mean systolic blood pressure (SBP), diastolic blood pressure (DBP), and mean arterial pressure (MAP) registered higher values than those observed before the onset of the pandemic. Furthermore, there was a notable rise in the percentage of participants with monthly average blood pressure readings classified as uncontrolled or severely uncontrolled arterial hypertension [68]. Nonetheless, it remains challenging to establish the underlying factors that contributed to this observed effect. It was not ascertained whether the individuals with elevated blood pressure had been infected with SARS-CoV-2. A similar investigative endeavor was carried out by the Excellence Centres of the European Society of Hypertension [69]. Their primary objective was to assess the influence of the COVID-19 pandemic on blood pressure control by conducting a comparative analysis of the results obtained through ABPM during the pandemic period in contrast to the pre-pandemic era.

### 2.5. No Changes in Blood Pressure Values after COVID-19

In contrast to the above-mentioned findings, the outcomes of the studies conducted by Nandadewa [66] and Sluijs [67] did not report any notable variations in blood pressure levels. A summary of the characteristics of these included studies can be found in Table 5. Nandadewa et al. [66] observed that ambulatory daytime, nighttime, and 24 h SBP, DBP, and mean blood pressure did not exhibit significant differences in the control and study groups. It is important to note that the transient effects of COVID-19, such as elevated blood pressure and central arterial stiffness, cannot be ruled out, especially when closer to the time of diagnosis. However, it is important to note that certain limitations, including a relatively small study population consisting of 28 young adults with COVID-19 and 10 control subjects, might impact the accuracy of the results.

Additionally, Sluijs [67] assessed the circulatory system and physical functioning in patients who did not require hospitalization during COVID-19 infection six months after their recovery. In their study, no significant changes in blood pressure were observed. The study by Jennifer K et al. has been discussed previously [55].

## 3. Discussion

The primary objective of this systematic review was to assess various studies examining the influence of SARS-CoV-2 on blood pressure elevation. The combined research suggests that the virus and the development of an infection may lead to an increase in blood pressure, potentially contributing to the onset of arterial hypertension. It is noteworthy that, among all the studies reviewed in this article, only three presented findings contradicting this hypothesis. However, the pathophysiological mechanisms potentially involved in the development of hypertension remain inadequately explored. The studies discussed in this review did not elucidate the specific mechanisms underlying the observed results. The researchers referred to previous studies on the impact of SARS-CoV-2 on vascular functions to comprehend the elements that might influence the observed outcomes. The potential factors contributing to blood pressure elevation after a COVID-19 infection are illustrated in Figure 1.

### 3.1. The Renin–Angiotensin–Aldosterone System, Inflammatory State, and Endothelial Damage

The renin–angiotensin–aldosterone system, inflammatory state, and endothelial damage are interconnected mechanisms that require simultaneous consideration. Inflammation has the potential to cause damage to the endothelium, disrupting the RAAS system. Additionally, the virus’s direct invasion into the endothelium through interaction with the ACE-2 receptor can disturb the balance in the RAAS system. Various systematic reviews have proposed these disruptions as potential explanations for the observed elevation in blood pressure.

Renin functions as an enzymatic catalyst in the conversion of angiotensinogen, a tetradecapeptide, into angiotensin I (AT-I), characterized by relatively weak biological activity. The integral angiotensin-converting enzyme (ACE), situated within the endothelial cell membrane, assumes a pivotal role in the transformation of AT-I into angiotensin II (AT-II). Subsequently, AT-II engages with its receptor, angiotensin II receptor type 1 (RAt-1), instigating a cascade of physiological responses encompassing vasoconstriction, prothrombotic effects, and the retention of water and sodium. Collectively, these orchestrated processes contribute to the regulation of optimal blood pressure and circulatory dynamics, particularly in instances of hypovolemia, and facilitate the restoration of fluid volume [70].

Another variant of ACE, known as angiotensin-converting enzyme type 2, operates by converting AT-II into angiotensin 1-7. This peptide acts on the MAS receptor and demonstrates a range of beneficial effects, including antifibrotic, antiproliferative, anti-inflammatory, vasodilatory, diuretic, and bradykinin–NO (nitric oxide) enhancing effects. The involvement of the bradykinin–NO pathway in this context is suggested by existing research findings [71,72].

AT-(1-7) functions as an antagonist to counteract the pressor effect of AT-I, consequently contributing to a decrease in blood pressure. It has been proposed that the absence of ACE-2 may result in elevated blood pressure and accelerate the progression of pressure overload-induced cardiac dysfunction [73].

The downregulation of ACE-2 disrupts the conversion of angiotensin I to angiotensin 1-9 (AT-1-9) and angiotensin II to angiotensin 1-7 (AT-1-7), leading to the intracellular accumulation of AT-II.

An additional physiological effect of angiotensin II involves the stimulation of aldosterone synthase. Aldosterone, in turn, induces an expansion of plasma volume and cardiac output by enhancing the reabsorption of sodium in the renal system [74].

ACE-2 and AT-II have a reciprocal regulatory relationship [75]. Ang II increases AT1R levels, which leads to phosphorylation and activation of metalloprotease 17 (ADAM17) expression. ADAM17, belonging to the disintegrin and metalloprotease family, facilitates the cleavage of membrane proteins, releasing them into the surrounding milieu through a process known as shedding. The shedding of ACE-2 induced by ADAM17 diminishes the protective effect of ACE-2 in hypertension. Interestingly, Ang II can undergo conversion by ACE-2 into Ang-(1-7) [76,77,78,79], thereby counteracting its own negative effects.

ACE-2 demonstrates the capability to enzymatically degrade apelin-13, a peptide implicated in vasoconstriction and the modulation of fluid balance. Furthermore, ACE-2 exhibits proficiency in degrading a spectrum of peptides unrelated to RAS, encompassing kynurenine, dynorphin A, and neurotensin [80]. However, when ACE-2 experiences excessive activity due to inappropriate stimulation or malfunctions, it may potentially contribute to pathological processes.

Recent findings indicate that ACE-2 overexpression also provides protection against neurogenic hypertension by modulating baroreceptor reflexes and autonomic function within the central nervous system (CNS) [81].

Consequently, ACE-2 plays a role in regulating blood pressure under physiological conditions, and a reduction in ACE-2 gene expression levels is associated with increased blood pressure [82]. A simplified schematic representation of angiotensinogen metabolism and the resulting effects of these transformations is presented in Figure 3.

Several studies propose that infection with SARS-CoV-2 may disrupt the equilibrium between ACE/ACE-2 and AT-II/angiotensin-(1-7), potentially influencing blood pressure and contributing to the onset of arterial hypertension [70]. In the context of SARS-CoV-2 infection, a notable decrease in ACE-2 expression has been observed, resulting in heightened activity of angiotensin II [83]. This augmented angiotensin II activity is implicated in vascular and organ damage. Ongoing investigations seek to elucidate the precise mechanisms by which SARS-CoV-2 impacts the renin–angiotensin–aldosterone system and blood pressure. A comprehensive understanding of the specific cardiovascular consequences associated with this infection remains an area of active research [84,85,86].

The initiation of SARS-CoV-2 infection is thought to occur in its host by attaching its structural protein, the S-glycoprotein, to angiotensin-converting enzyme 2 (ACE-2) [87,88,89,90]. A specific domain of the spike glycoprotein binds to the tip of subdomain I of ACE-2, facilitating fusion and supporting viral replication [91].

SARS-CoV-2 gains entry into host cells by utilizing the spike glycoprotein receptor. Following the fusion of the viral membrane with the host cell, viral RNA is released into the cytoplasm. Several transmembrane proteases, including ADAM17, transmembrane serine protease 2 (TMPRSS), and tumor necrosis factor (TNF)-converting enzyme, along with proteins like vimentin and clathrin are believed to play roles in the comprehensive process of establishing infection [92,93,94,95,96,97].

According to the research conducted by Wenhui et al. [87], SARS-CoV-2 demonstrated enhanced transfection efficiency in cells expressing ACE-2 in comparison to those lacking ACE-2. The findings additionally unveiled a distinct and high-affinity interaction between the S1 domain of the SARS-CoV-2 S protein and ACE-2 [98].

ACE-2 exhibits varying degrees of expression in nearly all human organs. During a SARS-CoV-2 infection, ACE-2 or its transmembrane domain is internalized alongside the virus. The subsequent release of soluble ACE-2 into the cellular supernatants is suggested to account for the substantial reduction in ACE-2 expression [83]. The downregulation of ACE-2 hampers the conversion of AT-I to AT-1-9 and AT-II to AT-1-7. Consequently, intracellular accumulation of AT-II occurs, promoting thrombosis by inducing the expression of plasminogen activator inhibitor-1 (PAI-1) in endothelial cells. PAI-1 inhibits tissue plasminogen activator (tPA) and urokinase plasminogen activator (uPA), leading to impaired fibrinolysis and an increased susceptibility to thrombosis [99].

Certain investigations propose that SARS-CoV-2 may augment the expression of disintegrin and metalloproteinase 17 (ADAM17) through interactions with ACE-2, leading to the subsequent release of ACE-2. This process results in impaired ACE-2 functionality, concurrently elevating the expression of ACE and angiotensin II. The heightened levels of angiotensin II induce the release of cytokines, including interleukins 6 (IL-6) and 7 (IL-7). These cytokines activate the mitogen-activated protein kinase (MAPK) pathway, thereby enhancing the expression of ADAM17 and establishing a positive feedback loop.

SARS-CoV-2, through its interaction with ACE-2, has been linked to various cardiovascular complications, including myocardial dysfunction, endothelial dysfunction, microvascular dysfunction, plaque instability, and myocardial infarction [100].

ACE-2 is a pivotal component of RAS, a critical regulator of blood pressure. The disruption of ACE-2 by COVID-19 remains a subject of investigation.

Researchers, including Xiong [38], Ozcan [42], Akpek [43], Fernandez-Ortega [46], Vyas [47], Gameil [62], and Mahmoud [65], suggested that the described interaction between COVID-19 and ACE-2 may elucidate their respective research findings. A subset of authors has attributed endothelial damage as the underlying cause for the interruption of the renin–angiotensin–aldosterone system (RAAS) (Shang [40], Delalic [44], Fernandez-Ortega [46], Vyas [47], Salon [63], Nandadeva [64]). Delic et al. [44] sought to identify predictive markers for blood pressure elevation resulting from COVID-19 infections, though none were extracted. Conversely, Ziyad Al-Aly [56] discovered that individuals with residual infection symptoms exhibited elevated levels of angiogenic, chemotactic, and post-inflammatory cytokines. Al-Aly proposed that immune system activation could be implicated in hypertension as a mechanism for post-acute sequelae of SARS-CoV-2 (PASC). Similarly, Zhang [51], Boglione [41], and Maestre-Muniz [49] have mentioned a dysregulated inflammatory response as a potential explanation for the heightened risk of hypertension. In contrast, Tanni et al. [61] did not observe a correlation between symptoms and inflammatory biomarkers.

### 3.2. Exacerbation of Hypertension Risk

The precise etiology of primary arterial hypertension remains elusive. However, several correlated risk factors, including obesity, chronic kidney diseases, metabolic disorders, a high-sodium diet, alcohol abuse, lack of physical activity, and inadequate sleep, show strong and independent associations with its development [101].

### 3.3. Psychological Impact

Hence, a hypothetical factor contributing to fluctuations in blood pressure levels may be the exacerbation of conventional risk factors associated with the development of hypertension. To some extent, this exacerbation may augment the mechanisms precipitating the onset of the condition. One such contributory factor is lipid disorders with hypercholesterolemia identified as an extensively studied risk factor for hypertension. Vyas [47] observed anomalies in lipid metabolism among individuals with established hypertension, although these changes did not attain statistical significance. Correspondingly, Ziyad Al-Aly [56] proposed that elevated blood pressure could result from both lipid metabolism disorders and diabetes that arise subsequent to a COVID-19 infection. Tisler [57] also noted an association between the development of diabetes and hypertension. In contrast, Abumayyleh [48] found that new onset hypertension was more prevalent in patients without diabetes, although the underlying pathomechanism for this correlation remains unexplained. Ogungbe [45] observed that COVID-19 side effects were more prevalent in older patients who were already categorized in the hypertension risk group. Consequently, a COVID-19 infection may potentially function as a risk multiplier and, consequently, an accelerator of hypertension occurrence. Cohen [52], Al-Aly [53], Mizrahi [54], Daugherty [50], and Ozecan [42] highlighted the occurrence of COVID-19 complications, such as diabetes or kidney, lipid, and sleep disorders, which are recognized as typical factors in hypertension development. It is established that an increase in blood pressure can be attributed to weight gain and fatigue, which are conditions that, in turn, promote a sedentary lifestyle—traits frequently associated with the pandemic and coronavirus infection. Van der Slujis [67] noted that the participants in their specific study were highly physically active. Given that physical activity is known to mitigate the risk of cardiovascular diseases, this could potentially account for the observed absence of blood pressure elevation in their research.

A recent meta-analysis has revealed a higher prevalence of depression, anxiety, insomnia, psychological distress, and post-traumatic stress disorder (PTSD) in populations affected by the pandemic [102]. Kaczynska et al. [103] suggested that both acute and chronic stress can contribute to the onset of hypertension in susceptible individuals.

In a prospective study involving over 160,000 healthy, non-obese adults aged 20 to 80 years, the occurrence of sleep durations less than 6 h per day was independently associated with an elevated risk of hypertension [104]. Therefore, an alternative avenue for exploring the elevation in blood pressure may be sought among psychological factors. The prevalence of anxiety about contracting the coronavirus during the pandemic was notably high, leading to consequential sleep disturbances [105]. Xiong [38], Mei [39], Boglione [41], Maestre-Muniz [49], and Tisler [57] proposed that the psychological state of patients might have influenced the study outcomes. However, it remains uncertain whether the observed blood pressure elevation is transient or could have enduring effects. Conversely, Akpek [43] emphasized that stress levels did not impact the results of his study, as it was conducted post-recovery when the patients’ mental states had significantly improved.

### 3.4. Autonomic Nervous System

The autonomic nervous system plays a crucial role in regulating perfusion in vital organs under diverse conditions. Arterial baroreceptors, located in the carotid sinus and aortic arch, relay information about blood pressure levels to the central nervous system [106]. In normal physiological conditions, activation of these baroreceptors in response to increased blood pressure stimulates vagal inhibitory neurons, reducing sympathetic neuron discharge, particularly in the peripheral vascular bed [107]. This process results in a decrease in heart rate, peripheral vascular resistance, and venous return, leading to a decline in blood pressure. Conversely, heightened activity of the sympathetic nervous system and inhibition of the vagus nerve lead to elevated blood pressure.

SARS-CoV-2 infections have been linked to inflammatory cytokine storms, and their penetration through the blood–brain barrier could potentially amplify the activation of the sympathetic nervous system (SNS). Therefore, there is a hypothesis that a COVID-19 infection may induce inflammation, damage, and dysfunction of baroreceptors [108]. The extent to which the vascular damage related to COVID-19 infection causes disorders in the baroreceptor reflex and contributes to autonomic system dysfunctions remains a significant area of research. Basic et al. [108], for instance, have identified characteristics of cardiac autonomic dysfunction in patients with COVID-19. Nevertheless, this topic remains relatively unexplored, possibly explaining its absence in most studies reviewed here that discuss potential mechanisms for hypertension.

However, Shang [40] and Salon [63] have proposed that persistent symptoms following a COVID-19 infection may be linked to compromised nervous system function and alterations in sympathetic nervous system activity. In contrast, Nandadewa [64], in a study challenging the notion that SARS-CoV-2 influences blood pressure elevation, underscores that transient increases in blood pressure may result from impaired vasodilation and the potential augmentation of vasodilation alongside heightened sympathetic nervous system activity.

## 4. Limitations

Recognizing the inherent limitations of this systematic review is crucial. Throughout our investigation, we identified only one meta-analysis by Zuin et al. [109] that addressed the risk of developing hypertension in individuals with COVID-19. They proposed that newly diagnosed hypertension could be a noteworthy complication of severe COVID-19 infection, citing five supporting studies within their review.

While our systematic review aimed to broaden the discussion on the influence of SARS-CoV-2 on blood pressure elevation, it is noteworthy that, despite discovering three studies contradicting the impact of COVID-19 on blood pressure, the majority of the analyzed studies align with the conclusions presented by Zuin et al. [109]. Given the limited availability of studies specifically delineating arterial hypertension as a complication of COVID-19 infection, we opted to include all studies demonstrating blood pressure elevation as a consequence of COVID-19 infection. It is acknowledged that this broader inclusion, covering studies describing various complications of SARS-CoV-2 infection, may introduce a potential bias towards affirming the thesis of this systematic review.

We would like to inform readers that the systematic review presented in this work was not registered in any registry before commencing data analysis. It is important to emphasize that, at the outset of the review, we were not fully acquainted with the practice of registering systematic reviews, its benefits, and associated procedures. Despite this limitation, our primary objective is to provide a comprehensive and valuable analysis within this review.

This systematic review is grounded in research obtained from two databases: Scopus and PubMed. Additionally, only studies published in English-language journals and those freely accessible in their entirety were considered, potentially resulting in the omission of some relevant publications. It is noteworthy that the study by Shegby et al. [110] was excluded from this review because it solely utilized blood pressure measurements to evaluate carotid artery and aortic stiffness in young adults with SARS-CoV-2. Nevertheless, these findings may indicate an increased risk of developing arterial hypertension.

Recognizing the limitations of the presented studies is essential, including factors like small study groups or the absence of control groups. In certain instances, the discussed studies lacked access to pre-infection patient observations, posing challenges in determining whether arterial hypertension pre-existed or was solely a consequence of SARS-CoV-2 infection.

## 5. Summary

Arterial hypertension constitutes a significant global health concern with an estimated doubling of affected adults from 1990 to 2019, now encompassing approximately 650 billion individuals [111,112]. The primary goal of treating arterial hypertension is to modify the risk factors, such as lipid disorders, obesity, diabetes, and smoking. Efforts to identify patients at risk of disease progression can facilitate the implementation of preventive measures, early detection, and complication avoidance [113,114].

Throughout the COVID-19 pandemic, considerable attention has been given to the impact of SARS-CoV-2 infection on various aspects of human physiology. Among the observed effects, the virus has been noted for its influence on blood pressure regulation with studies of COVID-19 patients revealing diverse effects on blood pressure levels. Even as the pandemic wanes, a persistent rise in COVID-19 infections underscores the ongoing necessity for research to comprehend the disease’s influence on blood pressure. Among the studies highlighted in this systematic review, only three included results contradicting the hypothesis of COVID-19 infection affecting blood pressure elevation. Strong arguments support the thesis that, through various mechanisms, COVID-19 infection can contribute to the development of hypertension. Nevertheless, we assert that these data hold significance within the realm of preventative measures and the assessment of patients post-COVID-19 recovery.

It is imperative to recognize the need for further and more comprehensive studies on this subject to better delineate the connections between increased blood pressure and COVID-19 infection.

## Figures and Tables

**Figure 1 ijms-25-01837-f001:**
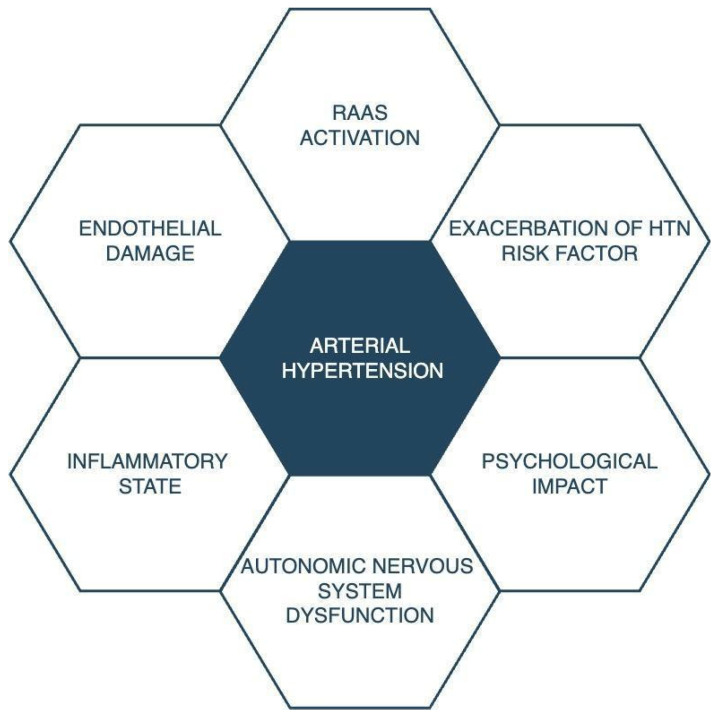
Possible impact of SARS-CoV-2 infection on the development of arterial hypertension. RAAS: renin–angiotensin–aldosterone system; HTN: arterial hypertension.

**Figure 2 ijms-25-01837-f002:**
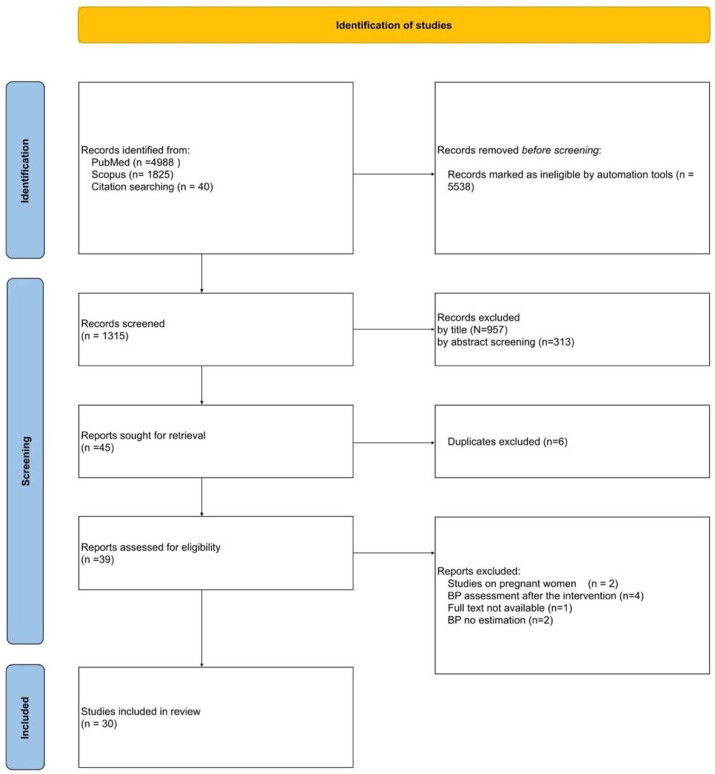
PRISMA flow diagram of the systematic review process performed in this study.

**Figure 3 ijms-25-01837-f003:**
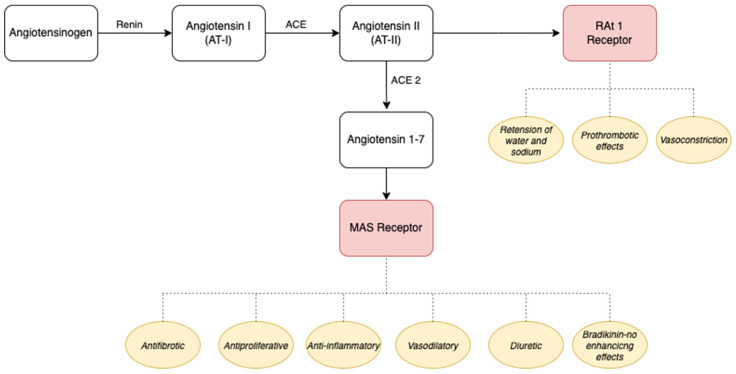
Angiotensinogen metabolism under the influence of ACE and ACE-2 together with the effects of end products. ACE-2: angiotensin-converting enzyme type 2; AT-I: angiotensin I; AT-II: angiotensin II; RAt-1 receptor: angiotensin Ⅱ receptor type 1.

**Table 1 ijms-25-01837-t001:** The qualitative evaluation and the qualitative criteria of the papers discussed for the systematic review.

Study/Quality Criteria	Study Design	Study Population	Control Group	Data Sources	Methods of Systematic Analysis	Reporting Statistical Significance	Ethics Committee Approval	Obtaining Informed Consent	Presenting Limitations
Xiong et al. [38]	+	+	+	+	+	+	+	-	+
Mei et al. [39]	+	+	-	+	+	+	+	+	+
Shang et al. [40]	+	+	-	+	+	+	+	-	+
Boglione et al. [41]	+	+	-	+	+	+	+	+	+
Ozcan et al. [42]	+	+	-	+	+	+	+	+	+
Akpek et al. [43]	+	+	-	+	+	+	+	-	+
Delalic et al. [44]	+	+	_	+	-	-	-	-	-
Ogungbe et al. [45]	+	+	-	+	-	+	+	+	+
Fernandez-Ortega MA [46]	+	+	-	+	-	-	+	+	+
Vyas et al. [47]	+	+	-	+	+	+	+	+	+
Abumayyaleh et al. [48]	+	+	-	+	+	+	+	-	+
Maestre-Muniz et al. [49]	+	+	-	+	+	+	+	-	+
Daugherty et al. [50]	+	+	+	+	+	+	+ *	-	+
Zhang HG et al. [51]	+	+	+	+	+	+	+	-	+
Cohen et al. [52]	+	+	+	+	+	+	+ *	-	+
Al-Aly et al. [53]	+	+	+	+	+	+	+	-	+
Mizrahi B et al. [54]	+	+	+	+	+	+	+	-	+
Jennifer K et al. [55]	+	+	+	+	+	+	+	- **	+
Ziyad Al-Aly et al. [56]	+	+	+	+	+	+	+	-	+
Tisler A et al. [57]	+	+	+	+	+	+	+	-	+
Alfadda et al. [58]	+	+	-	+	+	+	+	+	+
Tetiana et al. [59]	+	+	+	+	+	+	-	-	-
DeLorenzo et al. [60]	+	+	-	+	+	+	+	+	+
Tanni et al. [61]	+	+	-	+	-	-	-	-	+
Gameil et al. [62]	+	+	+	+	+	+	+	+	+
Saloň et al. [63]	+	+	+ ***	+	+	+	+	+	+
Nandadeva et al. [64]	+	+	+	+	+	+	+	+	-
Mahmoud et al. [65]	+	+	-	+	+	+	+	-	+
Nandadeva et al. [66]	+	+	+	+	+	+	+	+	-
vas der Sluijs et al. [67]	+	+	+	+	+	+	+	+	+

*+* data included, - no data. * This research was determined to be exempt from human research regulations by the UnitedHealth Group Office of Human Research Affairs. ** Informed consent was waived by the Helsinki committee. *** Lack of detailed information about the recruitment of the control group.

**Table 2 ijms-25-01837-t002:** Characteristics of studies with newly diagnosed arterial hypertension.

Study, (Year)	City/ Country	Sample Size	Disease Severity	Median/Mean Follow-Up Periods	Median/ Mean Age, % of Male in Case Group	% of Patients with Newly Diagnosed HTN	Obtain Data
Xiong et al. [38] (2020)	Wuhan, China	722	Mostly Severe, Critical	Median 97 days (95–102)	Median 52 (41–62) 45.5%	1.3	Telephone surveys
Mei et al. [39] (2021)	Wuhan, China	3677	Mild, Severe, Critical	Median 144 days (135–157)	Median 59 (47–68) 45.9%	0.16	Case, medical and self-reports
Shang et al. [40] (2021)	Wuhan, China	796	Severe Critical	6 months after infection	Median 62 (51–69) 50.8%	0.4	Telephone surveys
Boglione et al. [41] (2021)	Vercelli, Italy	449	Hospitalized	Median visit 1 32.5 days, visit 2 178.5 days	Median 65 (56–75.5) 78%	25.8—First visit 14—Second visit	Visit with examination
Ozcan et al. [42] (2022)	Turkey	406	Hospitalized	3 and 6 months	WHO-1: 46.8 ± 13.3 WHO-2 52.8 ± 13.1 WHO-3 54.8 ± 11.8	1	Telephone surveys
Akpek et al. [43] (2021)	Turkey	153	Mild	Mean 31.6 ± 5.0 days	Mean 46.5 ± 12.7 34%	11.76	Visit, examination
Delalic et al. [44] (2022)	Croatia, Zagreb	199	No data	Median 1 month	Mean age 57.3 46%	16.08	Visit, examination
Ogungbe et al. [45] (2022)	No data	442	Mild	Median 12.4 months (10–15.2)	Mean 45.4 29%	20	Telephone surveys
Fernandez-Ortega MA [46] (2023)	Mexico	70	Hospitalized	Follow up 5 months and 12 months	No data 65.7%	29.7 (5 months) 12.5 (12 months)	Telephone surveys
Vyas et al. [47] (2023)	India	248	Hospitalized	Follow-up length 1 year	Mean 51.16 ± 12.71 68.1%	32.3	Visit, examination
Abumayyaleh et al. [48] (2023)	International	3096	Severe	Follow-up time (months) diabetes 2.6 ± 4.6 non-diabetes 2.8 ± 4.9	Mean 72.6 ± 12.7 63.5%	DM patients 0.5% non-DM patients 1.6%	Telephone surveys
Maestre-Muniz et al. [49] (2021)	Spain	543	Hospitalized	12 months	Mean 65.1 (17.5; 18–98), 50.7%	2%	Telephone surveys

**Table 3 ijms-25-01837-t003:** Characteristics of included retrospective cohort studies using data from medical record databases.

Study, (Year)	City/ Country	Sample Size	Disease Severity	Median/Mean Follow-Up Periods	Median/Mean Age Małe %	HTN Risk
Daugherty et al. [50] (2021)	USA	9,247,505	Mild, Moderate, Severe	Median 87 days (45–124)	Mean 42.4 50.2%	risk ratio 1.81 (1.10 to 2.96)
Zhang HG et al. [51] (2022)	Germany, France, Italy, Singapore, USA	2,745,130	Hospitalized	Follow up 1 year	No data 74%	relative risk 1.14 (1.06–1.22)
Cohen et al. [52] (2022)	USA	2,895,943	Hospitalized	Median 78 days (30–175)	Mean 75.7 42%	risk difference 4.43 (2.27–6.37)
Al-Aly et al. [53] (2022)	USA	501,743	Mild, Moderate, Severe	Follow-up length 6 months	Mean 64.9 89.9%	hazard ratio 1.62
Mizrahi B et al. [54] (2023)	Israel	599,740	Mild	Two time periods after infection Early (30–180 days) Late (180–360 days)	Median 25 years old 49.4%	hazard ratio 1.27
Jennifer K et al. [55] (2022)	Israel	185,924	No data	Follow-up length: 14 months	No data	no difference
Ziyad Al-Aly et al. [56] (2021)	USA	5,064,270	Mild	Median 126 (81–203)	Mean 59.09 87.96%	hazard ratio 15.18 (11.53–18.62)
Tisler A et al. [57] (2022)	Estonia	19,460	Mild, Moderate, Severe	Mean 294.9	Mean 65.4 45.7%	hazard ratio 2.85

**Table 4 ijms-25-01837-t004:** Characteristics of the studies in which the blood pressure elevation as an outcome of COVID-19 infection was indicated.

Study, (Year)	City/ Country	Sample Size	Disease Severity	Median/Mean Follow-Up Periods	Median/Mean Age Male %	BP
Alfadda et al. [58] (2022)	Saudi Arabia	98	Hospitalized	Mean 7.02 ± 1.6 months	Mean 48.87 ± 17.11 51%	SBP mmHg 124.68 ± 14.9 vs. in follow-up 131.26 ± 15.3
Tetiana et al. [59] (2022)	Ukraine	115	Mild, Moderate	Mean 1.68 ± 1.2 months	Mean age 23.07 ± 1.54.	Patients with long COVID syndrome vs. control group SBP (127.1 ± 6.65 mmHg and 115.93 ± 6.24 mmHg and DBP 73.31 ± 5.30 mmHg vs. 68.79 ± 5.5 mmHg
DeLorenzo et al. [60] (2020)	Italy, Milan	185	Mild, Moderate, Severe	Median time from hospital discharge 23 days (20–29)	Mean age 57 male 66.5%	Uncontrolled BP requiring therapeutic change In 21.6% of patients
Tanni et al. [61] (2022)	Brazil	100	No data	Median 99 days	Mean age at 46.3. Mostly female	No data
Gameil et al. [62] (2021)	Egypt	240	Mild, Moderate	>3 months	Mean 38.29 55.9%	Control 120.63 ± 8.49 vs. research group 126.70 ± 10.31
Saloň et al. [63] (2023)	Austria	35	Hospitalized	Measurements day: 0/10 occurred 2 months after hospitalization.	Mean 60 ± 10 85%	142 mmHg to 150 mmHg
Nandadeva et al. [64] (2023)	USA, Texas	23	No Data	Median 15 months (3–30)	Mean 48 ± 9 0%	Systolic BP in COVID group 126 ± 19 vs. control: 109 ± 8 mmHg
Mahmoud et al. [65] (2022)	USA, Washington	100	Mild, Moderate, Severe	Median 99 days	Mean 46.3 19%	(Before COVID-19 disease vs. after) Median systolic BP 128 vs. 121.5 mmHg, median diastolic BP: 83.5 vs. 76 mmHg

**Table 5 ijms-25-01837-t005:** Characteristics of the studies in which no changes in BP values were observed.

Study, (Year)	City/ Country	Sample Size	Disease Severity	Median/Mean Follow Up Periods	Median/Mean Age Male %
Nandadeva et al. (2022) [66]	USA, Texas	38	Mild	Mean 11 ± 6 weeks	Control: 23 ± 3 yr COVID: 24.5 ± 4 yr 100%
vas der. Sluijs et al. (2022) [67]	The Netherlands	202	Mild	Median 175 days (126–235)	Mean 58 (54–65) 58%

## Data Availability

No new data were created or analyzed in this study. Data sharing is not applicable to this article.

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
