# Peer review of "Elevated Arterial Blood Pressure as a Delayed Complication Following COVID-19—A Narrative Review"

_ijms, 2024, doi:10.3390/ijms25031837_

Round 1

Reviewer 1 Report

Comments and Suggestions for Authors

Authors use the term hypertension. It should be precised what kind of hypertension they describe. In this case it is arterial hypertension.

This review is not well organised and comperhensively described. The authors just present the published research concerning the influence of COVID-19 on arterial hypertension but do not discuss sufficiently the pathomechanisms. The presented data is chaotic, the review should be thoroughly improved.

Comments on the Quality of English Language

The English language grammar should be thoroughly improved. It is not acceptable at present form.

Author Response

We appreciate the time and efforts in reviewing our manuscript. We have addressed all issues indicated in the review reports and believed that the revised version can meet the journal publication requirements.

Reviewer 1:

  • “Authors use the term hypertension. It should be precised what kind of hypertension they describe. In this case it is arterial hypertension.”

We precised the kind of hypertension as arterial hypertension.

  • This review is not well organised and comperhensively described. The authors just present the published research concerning the influence of COVID-19 on arterial hypertension but do not discuss sufficiently the pathomechanisms. The presented data is chaotic, the review should be thoroughly improved.

We completely reorganised and tried to comprehensively describe our review as well as we tried to discuss the potential pathomechanisms of the influence of COVID-19 on the blood pressure and arterial hypertension developing.

Reviewer 2 Report

Comments and Suggestions for Authors

The authors performed a comprehensive systematic review aiming to investigate the impact of COVID-19 infection on blood pressure and long-term risk of hypertension developing. Several mechanisms are appointed as a factor to increase blood pressure levels. Although the authors call this study as a systematic review, methodological aspects should be considered. Firstly, the study is not able to be reproducible with the information provided on methodology section. The authors did not cite PRISMA guidelines (or another) in the methods section. Crucial points were not considered in this review (e.g., characteristics of the population, study design). A better approach to understand the effects of COVID-19 on blood pressure levels would be done with direct comparisons between subjects previously infected with those were not. It is not possibly understood if the included studies were done to answer this question specifically. Furthermore, this is study is more a narrative than a systematic review a it should be considered. None Risk of Bias analysis were performed. For the appointed and several other factors, I cannot recommend this paper for publication.

Author Response

We appreciate the time and efforts in reviewing our manuscript. We have addressed all issues indicated in the review reports and believed that the revised version can meet the journal publication requirements.

Reviewer 2:

  • The authors performed a comprehensive systematic review aiming to investigate the impact of COVID-19 infection on blood pressure and long-term risk of hypertension developing. Several mechanisms are appointed as a factor to increase blood pressure levels. Although the authors call this study as a systematic review, methodological aspects should be considered. Firstly, the study is not able to be reproducible with the information provided on methodology section. The authors did not cite PRISMA guidelines (or another) in the methods section. Crucial points were not considered in this review (e.g., characteristics of the population, study design).

We completely modified our manuscript according to the PRISMA guidelines, especially the methods and results sections were changed. The population characteristics and the study design were added. We also made an effort to present all the limitations of our work, which we are aware.

  • A better approach to understand the effects of COVID-19 on blood pressure levels would be done with direct comparisons between subjects previously infected with those were not. It is not possibly understood if the included studies were done to answer this question specifically. Furthermore, this is study is more a narrative than a systematic review a it should be considered. None Risk of Bias analysis were performed.

We fully agree that the best way to understand the effects of COVID-19 on blood pressure levels would be done with direct comparisons between subjects previously infected with those were not. Unfortunately, we didn’t find such studies. We described studies which aimed the connection of COVID-19 with blood pressure. We tried to show the limitations of our work.

Reviewer 3 Report

Comments and Suggestions for Authors

In this review from Bielecka et al the authors assess the knowledge in regard to incidence of newly diagnosed hypertension cases in relation to COVID-19 infection.

Some comments:

Abstract is too general. Please specify how many articles supported increase in hypertension incidence after COVID-19 and how many argued against it.

another example: 'The possible role of the loss of angiotensin-converting enzyme 2, cytokine storm, increased sympathetic activity, and endothelial dysfunction were considered.' the context of this sentence is not present and therefore it does not say much.

Hoe the prothrombic effect of AT1 receptor (which authors state in section 1.2) exactly work? 

These two sentences are not supported by references. ''In the event of disruption to the circulatory system, angiotensin II is responsible for vasoconstriction and clotting at the site, followed by repair. However, excessive activity due to inappropriate stimulus or malfunction can lead to pathological processes such as hypertension or atherosclerosis.''

What does disruption of circulatory system mean? How does ANGII cause blood clotting?

'Another action of angiotensin II is stimulation of aldosterone synthase, a substance with a strong influence on electrolyte balance and a very strong antiproliferative effect.' This formulation of the sentence does not allow the understanding of the role of ANGII/Aldo in regulation of arterial pressure. Please rewrite.

Hoe does ACE2 facilitate a release of MMP17?

The last five sentences of the section 1.3 look like pasted without a structure. Please construct a sensible paragraph from those sentences. Otherwise, the just serve as separate statements.

Figure 2 is blurry and impossible to read.

There must be a table (or a detailed information in the text), which shows BP values in numbers. It is hard to follow when authors any speak about higher or elevated arterial pressure. After demonstrating these values, the authors should conform them to the classification of arterial blood pressure (both EU and USA).

Comments on the Quality of English Language

Many sentences need to be formulated in more understandable manner.

Author Response

We appreciate the time and efforts in reviewing our manuscript. We have addressed all issues indicated in the review reports and believed that the revised version can meet the journal publication requirements.

Reviewer 3:

  • Abstract is too general. Please specify how many articles supported increase in hypertension incidence after COVID-19 and how many argued against it. Another example: 'The possible role of the loss of angiotensin-converting enzyme 2, cytokine storm, increased sympathetic activity, and endothelial dysfunction were considered.' the context of this sentence is not present and therefore it does not say much.

The abstract was completely changed.

  • Hoe the prothrombic effect of AT1 receptor (which authors state in section 1.2) exactly work?

These two sentences are not supported by references. '

We added the explanation and the reference as following:

 The down regulation of ACE-2 impairs the conversion of Angiotensin I to Angiotensin 1-9 (AT-1-9) and Angiotensin II to Angiotensin 1-7 (A- 1-7), resulting in the intracellular accumulation of AT- II what promotes thrombosis by inducing the expression of plasminogen activator inhibitor-1 (PAI-1) in endothelial cells. PAI-1 inhibits tissue plasminogen activator (tPA) and urokinase plasminogen activator (uPA), results in hypofibrinolysis and thrombosis [41].

  • 'In the event of disruption to the circulatory system, angiotensin II is responsible for vasoconstriction and clotting at the site, followed by repair. However, excessive activity due to inappropriate stimulus or malfunction can lead to pathological processes such as hypertension or atherosclerosis.' What does disruption of circulatory system mean? How does ANGII cause blood clotting?

The indicated fragment was deleted during the manuscript editing and modification.

  • 'Another action of angiotensin II is stimulation of aldosterone synthase, a substance with a strong influence on electrolyte balance and a very strong antiproliferative effect.' This formulation of the sentence does not allow the understanding of the role of ANGII/Aldo in regulation of arterial pressure. Please rewrite.

The sentence was rewritten as following:

 Another action of angiotensin II is a stimulation of aldosterone synthase. Aldosterone lead to expansion of plasma volume and the cardiac output as a result of increase the reabsorption of sodium in the kidney  [42].

  • Hoe does ACE2 facilitate a release of MMP17?

The explanation was given in the manuscript as following:

Ang II increases AT1R levels, leads to phosphorylation and activation  metalloprotease 17 (ADAM17) expression. ADAM17 is a member of the disintegrin and metalloprotease family, cleaving membrane proteins and releasing them in the surrounding milieu through a process called shedding. ADAM17-induced ACE2 shedding reduce the protective effect of ACE-2 in hypertension, while Ang II can be converted by ACE-2 to Ang-(1-7) to inhibit its own negative effects.[44-47].

  • The last five sentences of the section 1.3 look like pasted without a structure. Please construct a sensible paragraph from those sentences. Otherwise, the just serve as separate statements.

The section 1.3 was completely changed according to suggestions.

  • Figure 2 is blurry and impossible to read.

Figure 2 was modified.

  • There must be a table (or a detailed information in the text), which shows BP values in numbers. It is hard to follow when authors any speak about higher or elevated arterial pressure. After demonstrating these values, the authors should conform them to the classification of arterial blood pressure (both EU and USA).

The Table 1 was divided into 4 to better represent the review results.

Table 3, called ” Characteristics of studies pointed at elevated blood pressure”, now presents BP values in numbers.

The information about classification of arterial blood pressure was added in the text.

Round 2

Reviewer 1 Report

Comments and Suggestions for Authors

Please consider changing the abbreviation HT to HTN, which is more common and widely used.

The figure 1 - please change hypertension to arterial hypertension

Line 405 - the sentence is not clear, please correct

The title of paragraph 3.5 should be changed

The introduction is too long. please reorganise.

The paragraph 4 which is entitled Discussion does not present the discussion but rather the summary. Please rename the title.

Please provide the discussion divided into paragraphs entitled according to the pathogenesis presented at Figure 1 .

Table 2: please correct the sample size column 

Table 3: please correct the title

All tables: please check carefully every column, many mistakes

Comments on the Quality of English Language

Moderate revisions required

Author Response

Thank you very much for all your comments. We do appreciate the time and efforts in reviewing our manuscript. We have addressed all issues indicated in the review report and believed that the revised version can meet the journal publication requirements.

Please consider changing the abbreviation HT to HTN, which is more common and widely used.
We changed the abbreviation according to your suggestion.

The figure 1 - please change hypertension to arterial hypertension
We changed it.

Line 405 - the sentence is not clear, please correct
We corrected the sentence. Now it is line 287.

The title of paragraph 3.5 should be changed
Title was changed.

The introduction is too long. please reorganise.
According to your suggestion, we reorganized the introduction and reduced its length.

The paragraph 4 which is entitled Discussion does not present the discussion but rather the summary. Please rename the title.
We renamed the title and a reorganized the text.

Please provide the discussion divided into paragraphs entitled according to the pathogenesis presented at Figure 1 .
We wrote a new discussion. We divided it into paragraphs according to the pathogenesis presented at Figure 1. To make it more attractive and understandable.

Table 2: please correct the sample size column.
All tables: please check carefully every column, many mistakes
All tables were carefully checked and corrected.

Table 3: please correct the title
We corrected the title.

Reviewer 2 Report

Comments and Suggestions for Authors

The work did not met criteria to be called as a systematic review. Several issues on methods previously reported were not accomplished. I don't recommend this article to publication, again!

Author Response

Thank you very much for your comments. We do appreciate the time and efforts in reviewing our manuscript. We have addressed all issues indicated in the review report and believed that the revised version can meet the journal publication requirements.

The work did not met criteria to be called as a systematic review. Several issues on methods previously reported were not accomplished. I don't recommend this article to publication, again!
We tried very hard to reorganize our paper to meet the standard of systemic review.

Round 3

Reviewer 1 Report

Comments and Suggestions for Authors

The authors made all relevant corrections.

Author Response

Dear Reviewer,

Thank you very much for your revision and approval.

Sincerely Yours,

Edyta Zbroch

Reviewer 2 Report

Comments and Suggestions for Authors

The work did not met criteria to be called as a systematic review. Several issues on methods previously reported were not accomplished. I don't recommend this article to publication, again!

Comments on the Quality of English Language

The work did not met criteria to be called as a systematic review. Several issues on methods previously reported were not accomplished. I don't recommend this article to publication, again!

Author Response

Dear Reviewer,

thank you again for your comments.

Please find the provided table with the qualitative evaluation and the qualitative criteria of the papers discussed for the systematic review in our manuscript.

We believe that the revised version can meet the journal publication requirements.

Regards,

Edyta Zbroch